# Reverse causation bias: A simulation study comparing first- and second-line treatments with an overlap of symptoms between treatment indication and studied outcome

Christian Bjerregård Øland[ID]*, Lise Skov Ranch, Tea Skaaby[ID], Thomas Delvin, Henny Bang Jakobsen, Christian Bressen Pipper[ID]

Biostatistics and Pharmacoepidemiology, LEO Pharma A/S, Ballerup, Denmark

* qbodk@leo-pharma.com

## Abstract

### Background

Reverse causation is a challenge in many drug-cancer associations, where the cancer symptoms are potentially mistaken for drug indication symptoms. However, tools to assess the magnitude of this type of bias are currently lacking. We used a simulation-based approach to investigate the impact of reverse causation on the association between the use of topical tacrolimus and cutaneous T-cell lymphoma (CTCL) in a multinational, population-based study using topical corticosteroids (TCS) as comparator.

### Methods

We used a multistate model to simulate patients' use over time of a first- (TCS) and second-line treatment (topical tacrolimus), onset of atopic dermatitis (indication for drugs) and CTCL (the studied outcome). We simulated different scenarios to mimic real-life use of the two treatments. In all scenarios, it was assumed that there was no causal effect of the first- or second-line treatment on the occurrence of CTCL. Simulated data were analysed using Cox proportional hazards models.

### Results

The simulated hazard ratios (HRs) of CTCL for patients treated with tacrolimus vs. TCS were consistently above 1 in all 9 settings in the main scenario. In our main analysis, we observed a median HR of 3.09 with 95% of the observed values between 2.11 and 4.69.

### Conclusions

We found substantial reverse causation bias in the simulated CTCL risk estimates for patients treated with tacrolimus vs. TCS. Reverse causation bias may result in a false positive association between the second-line treatment and the studied outcome, and this simulation-based framework can be adapted to quantify the potential reverse causation bias.

**Citation:** Øland CB, Ranch LS, Skaaby T, Delvin T, Jakobsen HB, Pipper CB (2024) Reverse causation bias: A simulation study comparing first- and second-line treatments with an overlap of symptoms between treatment indication and studied outcome. PLoS ONE 19(7): e0304145. https://doi.org/10.1371/journal.pone.0304145

**Data Availability Statement:** All relevant data are within the paper and its Supporting Information files.

**Funding:** This study was financially supported by LEO Pharma (https://www.leo-pharma.com/) in the form of salary for all authors and covering the cost associated with running the simulations. The specific roles of all authors are articulated in the 'author contributions' section. No additional external funding was received for this study. The funder had no role in study design, data collection and analysis, decision to publish, or preparation of the manuscript.

**Competing interests:** All authors are or have been employees of LEO Pharma A/S and may be shareholders of LEO Pharma A/S. LEO Pharma A/S owns and sells topical tacrolimus/Protopic®.

## Introduction

Reverse causation bias, also known as protopathic bias, arises in a drug-disease association if the drug under study is prescribed for symptoms of the outcome under study before this is correctly diagnosed. Reverse causation bias is a challenge in many drug-cancer associations, where the cancer symptoms are mistaken for drug indication symptoms, e.g., in the association of alpha-adrenoreceptor antagonists and prostate cancer;[1] respiratory drugs and lung cancer;[2] and proton pump inhibitors and colon cancer [1]. Studies of these may show a positive association between the drug and the specific cancer, even in the absence of a true association.

Another example is the association between topical tacrolimus, an ointment used to treat atopic dermatitis (AD), and cutaneous T-cell lymphoma (CTCL) [3]. AD is the most common type of eczema and is approximately 2,000 times more common than CTCL [4–6]. Since AD and CTCL may present similar symptoms, CTCL may initially be misdiagnosed as atopic dermatitis (AD) [3]. The first-line treatment for AD is topical corticosteroids (TCS) and the first-line treatment in early-stage CTCL is high-potency TCS [7]. If TCS cannot control the AD symptoms, second-line treatments (for example, topical tacrolimus) may be prescribed. In some cases, the delay from the first symptoms to the correct CTCL diagnosis may be years (for Mycosis fungoides, the most common type of CTCL, the median time from symptom onset to diagnosis was 3 to 4 years) [8, 9], during which the cutaneous symptoms are frequently treated as other skin conditions (for example, AD). Misclassification of CTCL as AD could give a false positive association between topical tacrolimus (second-line treatment) and CTCL [3], because patients with undiagnosed CTCL, misclassified as AD, are more likely to fail first-line treatment and receive second-line treatment for AD. This pattern could emerge in any comparison of first- and second-line treatments where there is an overlap in symptoms between indication and the outcome under study. Receiving the second-line treatment will appear associated with the outcome under study.

Reverse causation bias can sometimes be minimized by including a time-lag in the analysis where the cancer events that occur in a specified period after initiating medication (usually 1 or 2 years) are ignored or reclassified [2]. However, for diseases with overlapping symptoms where the time to correct diagnosis is longer than the commonly used lag time, this approach is not likely to solve the problem. On one hand, using a lag time shorter than the time to correct diagnosis, the approach is unlikely to sufficiently remove the reverse causation bias; on the other hand, extending the lag time even further increases the risk of removing any true drug-cancer association.

To our knowledge, the impact of reverse causation bias in studies comparing a second-line treatment to a first-line treatment with overlap in symptoms between the indication for the drug and the studied outcome has not been investigated. To investigate the potential impact of reverse causation bias in the association between the use of topical tacrolimus (second-line treatment) and CTCL in a large, multinational, population-based study [10], we used a simulation-based approach.

## Methods

### The JOELLE study

The Protopic® Joint European Longitudinal Lymphoma and Skin Cancer Evaluation (JOELLE) study was a population-based cohort study conducted in Denmark, Sweden, the Netherlands, and the United Kingdom to evaluate the risk of skin cancer and lymphoma associated with use of tacrolimus ointment [10]. This simulation study uses data from the JOELLE tacrolimus ointment cohorts (adults and children) and its matched TCS cohorts. The JOELLE

study used electronic health care data from 2002–2017: 32,605 children and 126,908 adult 'new' users of tacrolimus ointment matched to 117,592 children and 452,996 adult users of topical corticosteroids. Of the patients using tacrolimus ointment, 19.4% of adults and 31.9% children were followed for more than 10 years, and the median follow-up period was 5.0 years for adults and 5.7 years for children.

**The multistate model.**    In a survival analysis study, the patient can be simulated with two 'states' (alive and dead) and one possible transition (from being alive to dead). In this simulation study, the 'alive' state is partitioned into two or more intermediate (transient) states, each of which correspond to a stage of illness. Models linking progression through multiple states, also known as multistate models, can be used to simulate the transition of patients through the various states [11]. We used a multistate approach to simulate the transition of patients through the 'healthy', 'TCS-treated', 'tacrolimus-treated', and 'CTCL' states as illustrated in Fig 1.

All patients were born as 'healthy'. The patient later diagnosed with AD was assumed to be treated with either TCS or tacrolimus and moved to either of these states. If treated with TCS initially, the patient could switch treatment and progress to the tacrolimus state but not the opposite: patients stayed in the tacrolimus state until censoring or diagnosed with CTCL. This is in line with the JOELLE study. A diagnosis of CTCL was considered a terminal state from which no transition to other states occurred.

## Specification of time to event variables

*Age at onset of AD* was defined as the time from birth to the time of transition to one of the treatment states. We simulated *age at onset of AD* according to a distribution with a cumulative risk of AD of 6–7% at 18 years of age and 10–11% at 100 years of age (see Table 1 and S1 Fig for more details).

Features of this distribution were chosen to mimic the anticipated marginal *age at onset of AD* distribution in the general population from which patients were sampled for the JOELLE study.

*Age at onset of CTCL* was simulated assuming a constant incidence rate of 6 events per 1,000,000 person-years [12]. Following four million patients for 100 years or until death, whichever came first, we would expect approximately 1,574 CTCL cases. The proportion of

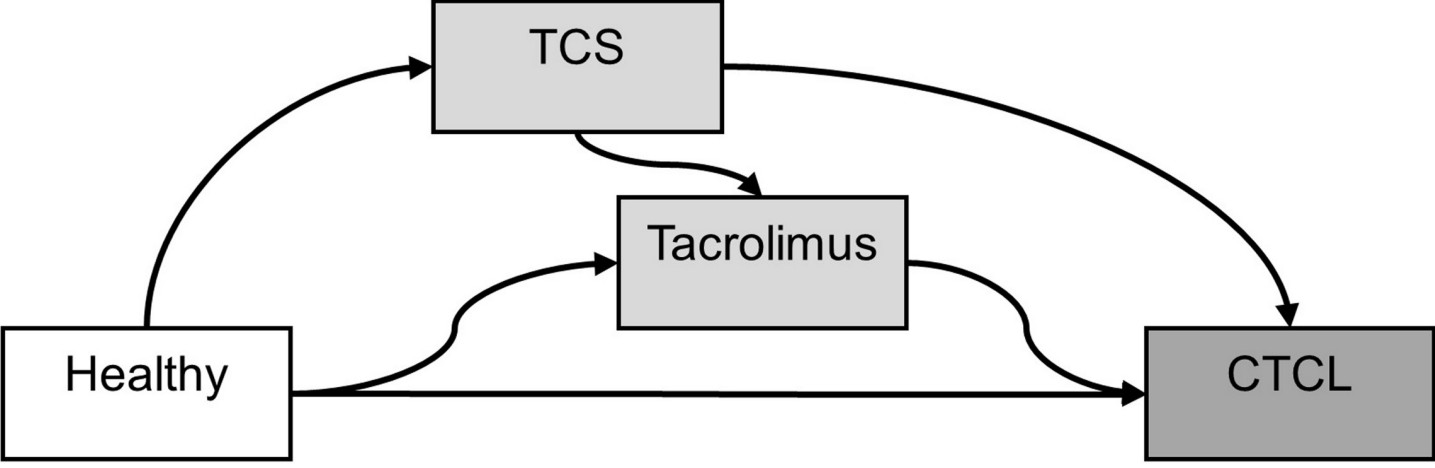

**Fig 1. Flowchart of the transitions in the multistate model applied in order to simulate data.** TCS: Topical corticosteroids; CTCL: cutaneous T-cell lymphoma.

**Table 1. Overview of variables used for the simulations.**

| Name | Definition | Description | Specification |
|---|---|---|---|
| Time from birth to AD. ($T_{AD}$) | $\frac{(-log(U_1))^4}{\lambda_{AD}}$ | Time from birth to AD in years. Corresponding approximately to a median age of onset at 5 years and 10% getting AD prior to year 100, not considering censoring. (See S1 Fig) | $\lambda_{AD} = 1.28 \ast 10^{-6}$<br>$U_1 \sim U(0,1)$ |
| Time from birth to CTCL. ($T_{CTCL}$) | $\frac{-log(U_2)}{\lambda_{CTCL}}$ | Time from birth to CTCL in years. Corresponding to an approximate median age of onset at 50 years, not considering censoring. | $\lambda_{CTCL} = 6 \ast 10^{-6}$<br>$U_2 \sim U(0,1)$ |
| Time from birth to censoring. ($T_{CENS}$) | $(1\text{-}exp(log(U_3)\ast2))\ast100$ | Time from birth to censoring in years. (See S2 Fig) | $U_3 \sim U(0,1)$ |
| Time from birth to diagnosis of AD. $T(_{AD,OBS})$ | $T_{AD}\ast I(T_{AD} < T_{CTCL}) + Q\ast$<br>$T_{AD}\ast I(T_{CTCL} \leq T_{AD}) + (1-$<br>$Q)\ast T_{CTCL}\ast I(T_{CTCL} \leq T_{AD})$ | Time from birth to diagnosis of AD in years. Q defines the proportion of patients being diagnosed correctly. | $Q \sim Bin(1,p_1)$ |
| Time from birth to TCS. ($T_{TCS}$) | $T_{AD,OBS}\ast S + T_{CENS}\ast(1-S)$ | Time from birth to TCS as first use in years. S defines the proportion of patients being treated with TCS as first-line treatment. | $S \sim Bin(1,p_2)$ |
| Time from birth to tacrolimus as first use. ($T_{TACRO,1}$) | $T_{AD,OBS}\ast(1-S) + T_{CENS}\ast S$ | Time from birth to tacrolimus as first use in years. | |
| Time from initiation of TCS to non-response to TCS. ($S_1$) | $(1 - exp(log(U_4)\ast2))\ast(1-W)+$<br>$(E_1 + 1)\ast W$ | Time from initiation of TCS to non-response to TCS treatment in years. (See S3 Fig) | $U_4 \sim U(0,1)$<br>$E_1 \sim EXP(3)$<br>$W \sim Bin(1,0.4)$ |
| Time from birth to tacrolimus. ($T_{TACRO}$) | $(((T_{TCS} + S_1)\ast P + T_{CENS}\ast \ (1 - P))\ast I(T_{TCS} = T_{AD}) +$<br>$((T_{TCS} + S_1)\ast R + T_{CENS}\ast(1-R))\ast I(T_{TCS} = T_{CTCL}))\ast S + T_{TACRO,1}\ast(1-S)$ | Time from birth to tacrolimus in years. P and R defines the proportion of AD and CTCL patients, respectively, switching to tacrolimus from TCS. | $P \sim Bin(1,p_3)$<br>$R \sim Bin(1,p_4)$ |
| Time from diagnosis of AD to re-evaluation of diagnosis. ($S_2$) | $max(wei(\lambda,\kappa,n))$ | Time from diagnosis of AD to re-evaluation of diagnosis in years. The Weibull distribution, $wei(\lambda,\kappa,n)$, with scale $\lambda$ and shape $\kappa$, the max of $n$ draws for each patient was picked. (See S4 Fig) | $\kappa = 0.9$<br>$n = 2$ |
| Time from birth to diagnosis of CTCL. $T(_{CTCL,OBS})$ | $T_{CTCL}\ast I(T_{AD,OBS} = T_{AD}) + (T_{AD,OBS} + S_2)\ast$<br>$I(T_{AD,OBS} = T_{CTCL})$ | Time from birth to diagnosis of CTCL in years. | |
| Status variable for TCS. | $I(T_{TCS} < T_{CENS})$ | Status variable (0/1) for treatment with TCS. | |
| Status variable for CTCL. | $I(T(_{CTCL,OBS}) < T_{CENS})$ | Status variable (0/1) for diagnosis of CTCL. | |
| Status variable for tacrolimus. | $I(T_{TACRO} < T_{CENS})$ | Status variable (0/1) for treatment with tacrolimus. | |

CTCL patients initially diagnosed correctly with CTCL was determined by a probability $p_1$ (Table 1).

*Age at diagnosis of AD* was defined as either the *age at onset of AD* or *age at onset of CTCL*. If *age at onset of AD* was prior to *age at onset of CTCL*, the patient was diagnosed with AD at *age at onset of AD*. Conversely, if *age at onset of CTCL* was prior to *age at onset of AD*, the

patient was either misdiagnosed with AD at *age at onset of CTCL* with probability 1-$p_1$, or correctly diagnosed with AD at *age at onset of AD* with probability $p_1$.

*The age at diagnosis of CTCL* was defined as either the *age at onset of CTCL* or, the time at which the correct CTCL diagnosis is given if the patient was misdiagnosed with AD at onset of CTCL. In the latter case, the time between AD diagnosis and CTCL diagnosis was simulated as the maximum of two independent draws from the same Weibull distribution: shape parameter was fixed at 0.9, and scale parameter ($\lambda$, the time taken to correctly diagnose CTCL) was varied according to different scenarios. In our main setting, $\lambda$ was set to 3.5 for a median duration of 4.4 years (inter-quartile range: 2.3–5.7 years) until CTCL was correctly diagnosed (S4 Fig).

*The follow-up period of the patient* mimicked the design of the JOELLE study. A 12-year study period was used: The patient was included in the cohort from a randomly chosen age (0–100 years) or age at AD diagnosis, whichever came last, and then followed for 12 years after the randomly chosen age or until diagnosis of CTCL or death, whichever came first. The distribution of age at death was chosen to reflect that of the general population (Netherlands, UK, Denmark, and Sweden) from which patients were sampled for the JOELLE study (see S2 Fig).

*The proportion of patients treated with TCS as first-line treatment* was defined as probability $p_2$. Note: 1-$p_2$ is the proportion of patients treated with tacrolimus as first-line treatment.

*The proportion of correctly diagnosed AD patients switching to treatment with tacrolimus* was defined as probability $p_3$; *the proportion of incorrectly diagnosed AD patients switching to treatment with tacrolimus* was defined as probability $p_4$. The time from initiation of TCS treatment to switching treatment was simulated in two steps: First step, we randomly selected 40% of the patients (the re-evaluation was assumed to be delayed for the remaining 60%) and simulated their duration using $(1 - exp(log(U_4)*2))$ ($U_4$ follows a uniform distribution in the range zero to one) to ensure that the patient had a duration on TCS before switching to tacrolimus below one year. Second step, for the remaining 60% of the patients, the duration was simulated according to an exponential distribution with rate 3 and shifted by 1 year (to ensure simulated durations above 1 year for these patients) (S3 Fig). $p_3$ and $p_4$ determined the magnitude and direction of the reverse causation bias. $p_3$ was low to reflect a good overall response to TCS and was varied to reflect the possibility of different methods of use in the four countries in the JOELLE study. $p_4$ was set lower than $p_3$ to reflect a lower response rate to TCS in patients with CTCL who were misdiagnosed with AD initially.

## Simulation specification

### Scenarios

In the simulations, the variables $p_1,p_2,p_3,p_4$, and $\lambda$ (governing the transition through states of the multi-state model) were varied to induce 3 scenarios: *main*, *maximum use*, and *discovery*.

In the *main scenario*, we assigned values to reflect the JOELLE study [10] (Table 2). It shows the predominant use of TCS as first line treatment ($p_2 = 0.95$) and few true AD patients switching to tacrolimus ($p_3 = 0.05$) from TCS. For misdiagnosed CTCL patients, the risk of switching to tacrolimus is higher ($p_4 = 0.10; 0.25; 0.40$), and the time to correct diagnosis is long ($\lambda = 3.5$).

In the *maximum use scenario*, we investigated a scenario where more patients used tacrolimus (to reflect potential country differences) by varying $p_2$, $p_3$, and $p_4$ (Table 3).

In the *discovery scenario*, we explored a scenario where misdiagnosed CTCL patients were diagnosed correctly with CTCL faster by varying $\lambda$ (Table 4).

**Table 2. Main scenario.**

| # | $p_1$ | $p_2$ | $p_3$ | $p_4$ | $\lambda$ | Rationale |
|---|---|---|---|---|---|---|
| A1 | 0.25 | 0.95 | 0.05 | 0.25 | 3.5 | Main assumption, based on observed JOELLE data, other available information and expert input. |
| - Increasing the proportion of correct CTCL diagnosis | | | | | | |
| A2 | 0.50 | 0.95 | 0.05 | 0.25 | 3.5 | The proportion of correctly CTCL diagnosed patients is assumed to be low but in testing this assumption we simulate 2 higher proportions. |
| A3 | 0.75 | 0.95 | 0.05 | 0.25 | 3.5 | |
| - Varying the proportion of first-line use of TCS | | | | | | |
| A4 | 0.25 | 0.99 | 0.05 | 0.25 | 3.5 | It is assumed that the number of patients who first receive TCS is at 95%, as sensitivity to this the assumption was tested at 99% and 90%. |
| A5 | 0.25 | 0.90 | 0.05 | 0.25 | 3.5 | |
| - Varying the proportion of AD patients switching to tacrolimus | | | | | | |
| A6 | 0.25 | 0.95 | 0.10 | 0.25 | 3.5 | The assumed proportion of patients switch to tacrolimus is assumed to be low due to it being a second-line treatment. This assumption was tested against 2 higher proportions. |
| A7 | 0.25 | 0.95 | 0.20 | 0.25 | 3.5 | |
| - Varying the proportion of (non-diagnosed) CTCL patients switching to tacrolimus | | | | | | |
| A8 | 0.25 | 0.95 | 0.05 | 0.10 | 3.5 | The assumed proportion of CTCL patients switching to tacrolimus is assumed to be at 25% reflecting a naturally lower response rate to TCS. This assumption was tested against a lower and a higher assumption. |
| A9 | 0.25 | 0.95 | 0.05 | 0.40 | 3.5 | |

$p_1$ = proportion of correct CTCL diagnosis; $p_2$ = proportion of first-line use of TCS; $p_3$ = proportion of AD patients switching to tacrolimus; $p_4$ = proportion of misdiagnosed CTCL patients switching to tacrolimus; $\lambda$ = scale parameter of the Weibull distribution used in the time to re-evaluation variable.

## Settings

We evaluated 9 different *settings* in the *main scenario*, 3 in the *maximum use scenario*, and 2 in the *discovery scenario*. For each setting, 1,000 replicate AD cohorts were simulated. Each replicate AD cohort was sampled as the subset of patients that were diagnosed with AD prior to or during follow-up from a simulated base population of 4 million patients. The resulting cohort size is approximately equal to the adult cohort in the JOELLE study.

## Statistical analyses

Data was simulated and analysed using the open-source statistical software, R (v. 3.6.1) and the add-on packages "mstate" (v. 0.2.12) and "survival" (v. 3.1–8) [13, 14]. Code used for the simulations and their analysis can be found in S1 File, along with the output of each simulation. Statistical analyses were carried out using Cox regression and Poisson regression.

**Table 3. Maximum use scenario.**

| # | $p_1$ | $p_2$ | $p_3$ | $p_4$ | $\lambda$ | Rationale |
|---|---|---|---|---|---|---|
| B1 | 0.25 | 0.90 | 0.20 | 0.40 | 3.5 | Sensitivity assumption, testing what happens if tacrolimus-use is more prevalent. |
| - Increasing the proportion of correct CTCL diagnosis | | | | | | |
| B2 | 0.50 | 0.90 | 0.20 | 0.40 | 3.5 | The proportion of correctly CTCL diagnosed patients is assumed to be low (also in this sensitivity assumption) but in testing this assumption we simulate 2 higher proportions. |
| B3 | 0.75 | 0.90 | 0.20 | 0.40 | 3.5 | |

$p_1$ = proportion of correct CTCL diagnosis; $p_2$ = proportion of first-line use of TCS; $p_3$ = proportion of AD patients switching to tacrolimus; $p_4$ = proportion of misdiagnosed CTCL patients switching to tacrolimus; $\lambda$ = scale parameter of the Weibull distribution used in the time to re-evaluation variable.

**Table 4. Discovery scenario.**

| # | p₁ | p₂ | p₃ | p₄ | λ | Rationale |
|---|---|---|---|---|---|---|
| C1 | 0.25 | 0.95 | 0.05 | 0.25 | 1.2 | The main assumption's guess of when a correct re-diagnosis of CTCL will happen in relation to patients who were mis-diagnosed as AD patients is tested here. 2 assumptions were tested, assuming earlier discovery. |
| C2 | 0.25 | 0.95 | 0.05 | 0.25 | 0.5 | |

$p_1$ = proportion of correct CTCL diagnosis; $p_2$ = proportion of first-line use of TCS; $p_3$ = proportion of AD patients switching to tacrolimus; $p_4$ = proportion of misdiagnosed CTCL patients switching to tacrolimus; λ = scale parameter of the Weibull distribution used in the time to re-evaluation variable.

Tacrolimus was categorised according to the ever-use principle in the JOELLE study: [10] once the patient had received tacrolimus, they would add risk-time to the tacrolimus cohort until end of follow-up. If the patient was prescribed TCS before tacrolimus, the patient would add risk time to the TCS cohort until first tacrolimus use. To quantify the magnitude and direction of the potential bias due to reverse causation, cohorts were simulated assuming no causal effect of TCS or tacrolimus on onset of CTCL, analysing only patients with diagnosed AD.

For each simulated cohort, we used Cox regression to analyse time from AD treatment initiation to CTCL (accounting for left truncation and right censoring due to follow-up period or death) and including a time dependent treatment variable in the regression. The resulting hazard ratio (the relative change in hazard rate after start of tacrolimus treatment compared to TCS treatment) was estimated by maximizing the Cox partial likelihood [15]. Treatment-specific incidence rates were estimated by Poisson regression with a log link (including a treatment variable in the regression) and using log person years at risk within each treatment group as offset. Within each simulation setting, the sample distribution of point estimates from the 1,000 simulations is shown by a histogram and numerically by median and a 95% reference interval (RI) defined as the range from the 2.5% to the 97.5% percentile.

## Results

The 3 simulation scenarios (*main scenario*, *maximum use scenario* and *discovery scenario*) assumed no causal effect of tacrolimus vs. TCS treatment on CTCL, that means any deviations from 1 of the simulated hazard ratios would indicate bias due to reverse causation.

### Main scenario

The simulated estimated hazard ratios of CTCL for patients treated with tacrolimus vs. TCS were above 1 in all 9 settings (Table 5).

For the primary setting (setting A1: characterised by a low chance of correctly diagnosing CTCL initially [$p_1 = 0.25$]), simulations showed a median estimated incidence rate of 0.145 (95% RI: 0.102, 0.205) CTCL cases per 1,000 person years during tacrolimus treatment and 0.047 (95% RI: 0.038, 0.056) CTCL cases per 1,000 person years during TCS treatment (Table 6).

The corresponding median hazard ratio of CTCL was 3.09 (95% RI: 2.11, 4.69) (Table 5 and Fig 2) for tacrolimus vs. TCS. In 99.7% of the simulations in this setting there was evidence in favour of a hazard ratio above 1.

Table 7 contains information explaining settings A1-9.

When increasing the likelihood of correctly diagnosing CTCL (settings A2 and A3 with $p_1 = 0.5$ and $p_1 = 0.75$, respectively), the simulated hazard ratios were closer to 1 but still revealed a substantial bias (Table 5 and Fig 2). When decreasing the use of TCS as the first-line

**Table 5. Results: Summary of estimated hazard ratios.**

| # | Variables | | | | | HR | 95% RI | min;max | %[1] |
|---|---|---|---|---|---|---|---|---|---|
| | $p_1$ | $p_2$ | $p_3$ | $p_4$ | $\lambda$ | | | | |
| A1 | 0.25 | 0.95 | 0.05 | 0.25 | 3.5 | 3.09 | (2.11;4.48) | (1.29;6.52) | 99.7 |
| A2 | 0.50 | 0.95 | 0.05 | 0.25 | 3.5 | 3.00 | (1.75;4.69) | (1.28;5.68) | 97.3 |
| A3 | 0.75 | 0.95 | 0.05 | 0.25 | 3.5 | 2.65 | (1.25;4.80) | (0.39;6.30) | 78.9 |
| A4 | 0.25 | 0.99 | 0.05 | 0.25 | 3.5 | 4.49 | (2.82;6.68) | (2.20;8.45) | 100.0 |
| A5 | 0.25 | 0.90 | 0.05 | 0.25 | 3.5 | 2.38 | (1.64;3.36) | (1.24;4.40) | 98.3 |
| A6 | 0.25 | 0.95 | 0.10 | 0.25 | 3.5 | 2.03 | (1.32;2.89) | (0.87;3.91) | 90.4 |
| A7 | 0.25 | 0.95 | 0.20 | 0.25 | 3.5 | 1.14 | (0.75;1.69) | (0.55;2.40) | 12.4 |
| A8 | 0.25 | 0.95 | 0.05 | 0.10 | 3.5 | 1.41 | (0.79;2.30) | (0.41;3.06) | 30.1 |
| A9 | 0.25 | 0.95 | 0.05 | 0.40 | 3.5 | 5.43 | (3.63;7.78) | (2.89;9.80) | 100.0 |
| B1 | 0.25 | 0.90 | 0.20 | 0.40 | 3.5 | 1.86 | (1.32;2.59) | (1.15;3.40) | 93.4 |
| B2 | 0.50 | 0.90 | 0.20 | 0.40 | 3.5 | 1.80 | (1.11;2.76) | (0.91;3.41) | 76.8 |
| B3 | 0.75 | 0.90 | 0.20 | 0.40 | 3.5 | 1.66 | (0.93;2.93) | (0.51;4.11) | 45.8 |
| C1 | 0.25 | 0.95 | 0.05 | 0.25 | 1.2 | 2.39 | (1.49;3.54) | (0.86;4.33) | 94.7 |
| C2 | 0.25 | 0.95 | 0.05 | 0.25 | 0.5 | 1.44 | (0.80;2.28) | (0.50;3.83) | 33.1 |

$p_1$ = proportion of correct CTCL diagnosis; $p_2$ = proportion of first-line use of TCS; $p_3$ = proportion of AD patients switching to tacrolimus; $p_4$ = proportion of CTCL patients switching to tacrolimus; $\lambda$ = scale parameter of the Weibull distribution used in the time to re-evaluation variable; RI = Reference interval; HR = median hazard ratio.

[1]: Risk of falsely concluding a treatment effect, defined as the proportion of simulations providing a statistically significant result.

treatment (setting A5, $p_2$ = 0.90), the bias was also reduced. Conversely, when increasing the use of TCS as the first-line treatment (setting A4, $p_2$ = 0.99), the simulated hazard ratio and thus the bias was substantially higher.

**Table 6. Summary of estimated incidence rates.**

| # | Variables | | | | | Tacrolimus | | | TCS | | |
|---|---|---|---|---|---|---|---|---|---|---|---|
| | $p_1$ | $p_2$ | $p_3$ | $p_4$ | $\lambda$ | IR | 95% RI | min;max | IR | 95% RI | min;max |
| A1 | 0.25 | 0.95 | 0.05 | 0.25 | 3.5 | 0.145 | (0.102;0.205) | (0.071;0.248) | 0.047 | (0.038;0.056) | (0.030;0.063) |
| A2 | 0.50 | 0.95 | 0.05 | 0.25 | 3.5 | 0.098 | (0.062;0.142) | (0.044;0.177) | 0.033 | (0.026;0.041) | (0.020;0.049) |
| A3 | 0.75 | 0.95 | 0.05 | 0.25 | 3.5 | 0.053 | (0.026;0.085) | (0.009;0.120) | 0.020 | (0.014;0.026) | (0.010;0.030) |
| A4 | 0.25 | 0.99 | 0.05 | 0.25 | 3.5 | 0.208 | (0.136;0.288) | (0.104;0.388) | 0.047 | (0.038;0.056) | (0.033;0.061) |
| A5 | 0.25 | 0.90 | 0.05 | 0.25 | 3.5 | 0.111 | (0.077;0.149) | (0.062;0.173) | 0.047 | (0.038;0.057) | (0.034;0.062) |
| A6 | 0.25 | 0.95 | 0.10 | 0.25 | 3.5 | 0.099 | (0.068;0.137) | (0.051;0.160) | 0.049 | (0.039;0.059) | (0.033;0.065) |
| A7 | 0.25 | 0.95 | 0.20 | 0.25 | 3.5 | 0.062 | (0.043;0.085) | (0.033;0.102) | 0.054 | (0.044;0.066) | (0.036;0.073) |
| A8 | 0.25 | 0.95 | 0.05 | 0.10 | 3.5 | 0.076 | (0.044;0.116) | (0.022;0.169) | 0.054 | (0.044;0.064) | (0.037;0.072) |
| A9 | 0.25 | 0.95 | 0.05 | 0.40 | 3.5 | 0.216 | (0.155;0.278) | (0.120;0.328) | 0.039 | (0.031;0.048) | (0.027;0.053) |
| B1 | 0.25 | 0.90 | 0.20 | 0.40 | 3.5 | 0.084 | (0.065;0.107) | (0.054;0.119) | 0.045 | (0.036;0.056) | (0.029;0.063) |
| B2 | 0.50 | 0.90 | 0.20 | 0.40 | 3.5 | 0.058 | (0.040;0.077) | (0.029;0.100) | 0.032 | (0.024;0.042) | (0.021;0.051) |
| B3 | 0.75 | 0.90 | 0.20 | 0.40 | 3.5 | 0.032 | (0.020;0.048) | (0.009;0.062) | 0.019 | (0.013;0.026) | (0.009;0.031) |
| C1 | 0.25 | 0.95 | 0.05 | 0.25 | 1.2 | 0.123 | (0.080;0.170) | (0.044;0.192) | 0.051 | (0.042;0.060) | (0.037;0.065) |
| C2 | 0.25 | 0.95 | 0.05 | 0.25 | 0.5 | 0.080 | (0.044;0.116) | (0.031;0.146) | 0.055 | (0.045;0.066) | (0.038;0.073) |

$p_1$ = proportion of correct CTCL diagnosis; $p_2$ = proportion of first-line use of TCS; $p_3$ = proportion of AD patients switching to tacrolimus; $p_4$ = proportion of CTCL patients switching to tacrolimus; $\lambda$ = scale parameter of the Weibull distribution used in the time to re-evaluation variable; RI = reference interval; IR = median incidence rate. Incidence rates are presented as number of events per 1000 person-years

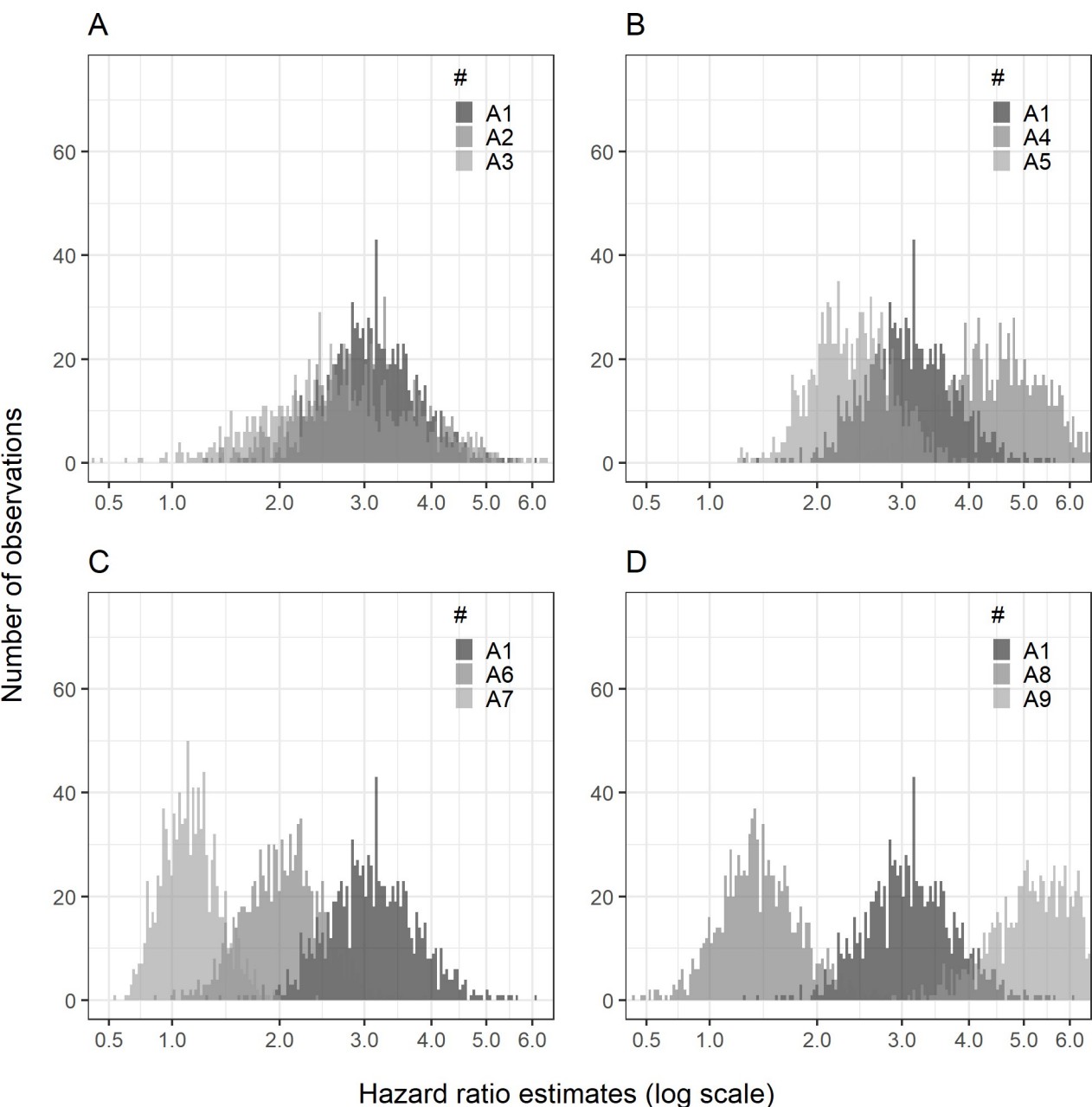

**Fig 2. Main scenario histogram.**

When decreasing the difference in likelihood of switching to tacrolimus between true AD patients and misdiagnosed CTCL patients either by increasing the likelihood of switching for true AD patients (setting A6 and A7, $p_3$ = 0.10, 0.20; $p_4$ = 0.25) or by decreasing the likelihood of switching for misdiagnosed CTCL patients (setting A8, $p_3$ = 0.05, $p_4$ = 0.10) the resulting bias is still present but lower. When increasing the difference (setting A9), the simulated bias was markedly larger (Table 5 and Fig 2).

**Table 7. Main scenario legend.**

|  | Main scenario | Increasing the proportion of correct CTCL diagnosis | | Varying the proportion of first-line use of TCS | | Varying the proportion of AD patients switching to tacrolimus | | Varying the proportion of CTCL patients switching to tacrolimus | |
| --- | --- | --- | --- | --- | --- | --- | --- | --- | --- |
| # | A1 | A2 | A3 | A4 | A5 | A6 | A7 | A8 | A9 |
| $p_1$ | 0.25 | 0.50 | 0.75 | 0.25 | 0.25 | 0.25 | 0.25 | 0.25 | 0.25 |
| $p_2$ | 0.95 | 0.95 | 0.95 | 0.99 | 0.90 | 0.95 | 0.95 | 0.95 | 0.95 |
| $p_3$ | 0.05 | 0.05 | 0.05 | 0.05 | 0.05 | 0.10 | 0.20 | 0.05 | 0.05 |
| $p_4$ | 0.25 | 0.25 | 0.25 | 0.25 | 0.25 | 0.25 | 0.25 | 0.10 | 0.40 |
| $\lambda$ | 3.5 | 3.5 | 3.5 | 3.5 | 3.5 | 3.5 | 3.5 | 3.5 | 3.5 |

$p_1$ = proportion of correct CTCL diagnosis; $p_2$ = proportion of first-line use of TCS; $p_3$ = proportion of AD patients switching to tacrolimus; $p_4$ = proportion of CTCL patients switching to tacrolimus; $\lambda$ = scale parameter of the Weibull distribution used in the time to re-evaluation variable. Log-transformed x-axis of the hazard ratio estimates

## Maximum use scenario

In the *maximum use scenario*, we considered three settings with more use of tacrolimus compared to the *main scenario* by applying two actions: (1) increasing the frequency of first-line use of tacrolimus by decreasing $p_2$; and (2) increasing the frequency of second-line use of tacrolimus in true AD patients by increasing $p_3$ and simultaneously increasing the frequency of second-line use of tacrolimus in misdiagnosed CTCL patients by increasing $p_4$.

Applying both actions (1) and (2) by doubling the use of tacrolimus both as first-line and second-line treatment (setting B1) compared to the primary setting of the *main scenario*, the median hazard ratio was 1.86 (95% RI: 1.32, 2.59) (Table 5 and Fig 3).

Table 8 contains information explaining settings B1-9.

If incorrectly interpreting the hazard ratio as an increased risk of CTCL due to tacrolimus treatment compared to TCS treatment, we would declare a false positive association in 93.4% of the 1,000 simulations (Table 6). Although less pronounced than the primary setting of the *main scenario*, the bias induced by reverse causation in this setting is still substantial.

If, in addition to the doubling of tacrolimus treatment, we increase the chance of correctly diagnosing CTCL initially (settings B2: $p_1 = 0.5$; B3: $p_1 = 0.75$), the bias is further reduced (for example, the median hazard ratio is 1.66 [95% RI: 0.93–2.93] in setting B3 [$p_1 = 0.75$]) (Table 5 and Fig 3).

## Discovery scenario

In the *discovery scenario*, the expected time to a correct CTCL diagnosis is decreased (settings C1: $\lambda = 1.2$; C2: $\lambda = 0.5$) and thus CTCL patients are less likely to initiate second-line treatment with tacrolimus. Simulation results from this scenario show hazard ratios closer to 1, that is a decreased bias (Table 5 and Fig 4).

Table 9 contains information explaining settings A1, C1, and C2.

## Discussion

We investigated the association between the use of topical tacrolimus (second-line treatment) and CTCL using TCS (first-line treatment) as the comparator in a setting that assumed no association between treatment and CTCL. In the simulations, the calculated CTCL risk estimates for patients treated with tacrolimus vs. TCS, the reverse causation bias was larger when:

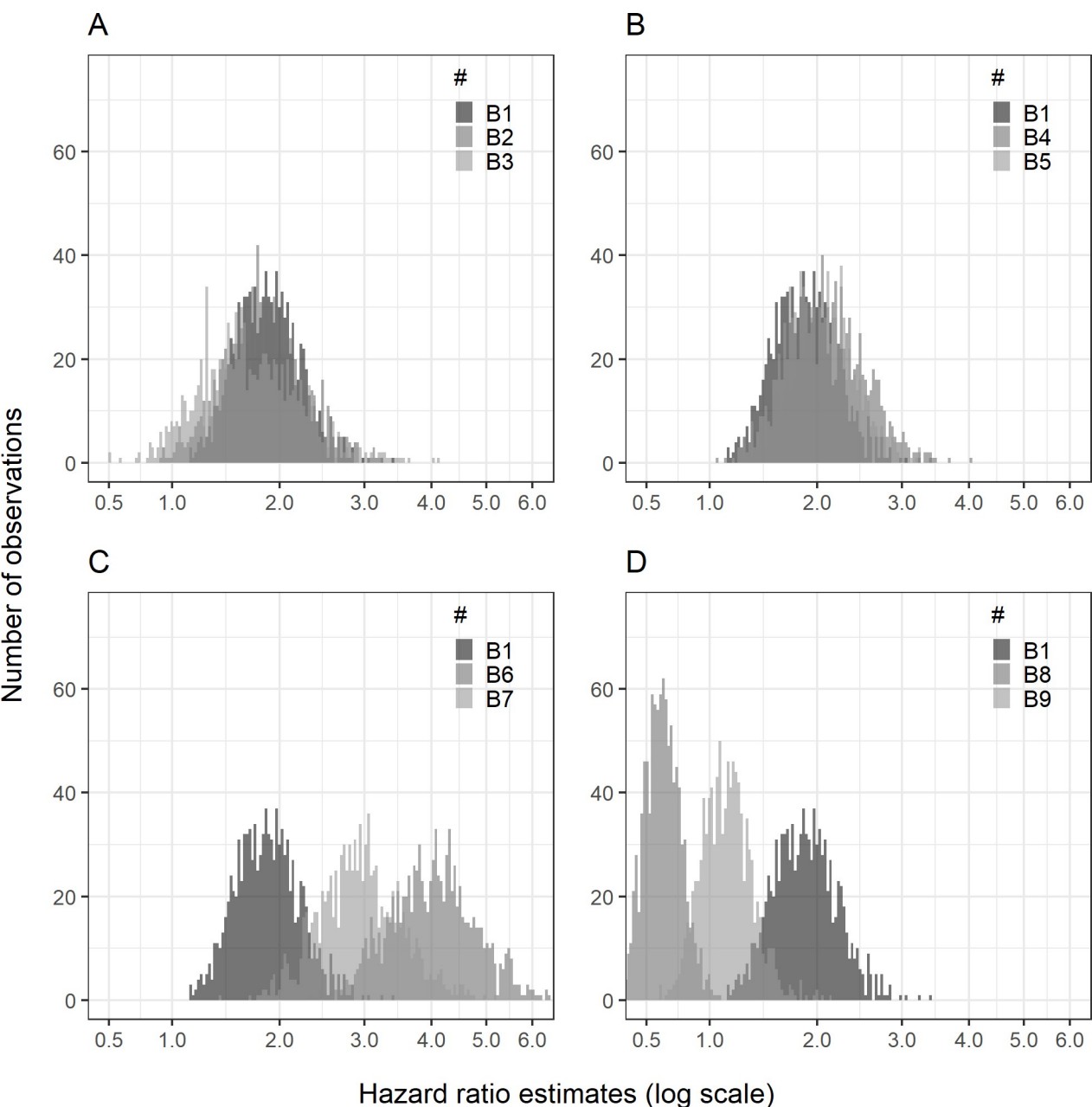

**Fig 3. Maximum use scenario histogram.**

1) increasing the use of the first line treatment; 2) increasing the difference in likelihood of switching to second-line treatment for true patients and misdiagnosed patients; 3) decreasing the chance of correctly diagnosing patients with the event of interest initially, thus increasing the number of misdiagnosed patients; 4) increasing the expected duration until correct diagnosis of the event of interest; and 5) decreasing the general use of second-line treatment, and vice versa.

The difference in the likelihood of switching to the second-line treatment governs the difference in the perceived risk profiles for the two patient groups. A first-line versus second-line treatment strategy in which the chance of switching to the second-line treatment is higher

**Table 8. Maximum use scenario legend.**

| # | Maximum use scenario | Increasing the proportion of correct CTCL diagnosis | | Varying the proportion of first-line use of TCS | | Varying the proportion of AD patients switching to tacrolimus | | Varying the proportion of CTCL patients switching to tacrolimus | |
|---|---|---|---|---|---|---|---|---|---|
| | B1 | B2 | B3 | B4 | B5 | B6 | B7 | B8 | B9 |
| $p_1$ | 0.25 | 0.50 | 0.75 | 0.25 | 0.25 | 0.25 | 0.25 | 0.25 | 0.25 |
| $p_2$ | 0.90 | 0.90 | 0.90 | 0.99 | 0.95 | 0.90 | 0.90 | 0.90 | 0.90 |
| $p_3$ | 0.20 | 0.20 | 0.20 | 0.20 | 0.20 | 0.05 | 0.10 | 0.20 | 0.20 |
| $p_4$ | 0.40 | 0.40 | 0.40 | 0.40 | 0.40 | 0.40 | 0.40 | 0.10 | 0.25 |
| $\lambda$ | 3.5 | 3.5 | 3.5 | 3.5 | 3.5 | 3.5 | 3.5 | 3.5 | 3.5 |

$p_1$ = proportion of correct CTCL diagnosis; $p_2$ = proportion of first-line use of TCS; $p_3$ = proportion of AD patients switching to tacrolimus; $p_4$ = proportion of CTCL patients switching to tacrolimus; $\lambda$ = scale parameter of the Weibull distribution used in the time to re-evaluation variable. Log-transformed x-axis of the hazard ratio estimates.

with more refractory symptoms in misdiagnosed CTCL patients, reverse causation is induced. Indeed, when switching to second-line treatment becomes more likely for misdiagnosed patients than true patients, reverse causation is induced.

Specifically, we observed an increase in bias with increased difference in likelihood of switching to tacrolimus between true AD patients and misdiagnosed CTCL patients. The bias dictated a higher risk of diagnosis of CTCL in tacrolimus users only because such users are more likely to have been misdiagnosed with AD instead of CTCL initially.

The average incidence rates of CTCL during TCS treatment vary from 0.019 to 0.063 cases per 1,000 person-years in our simulations, and are comparable to the estimated crude incidence rate of CTCL for adults during TCS treatment (0.04 [95% CI: 0.032, 0.05] per 1,000 person-years) in the JOELLE study [10]. In the adult UK cohort of the JOELLE study, the crude and adjusted incidence rate ratios and 95% CI were 3.11 (95% CI: 1.38, 6.91) and 3.12 (95% CI:

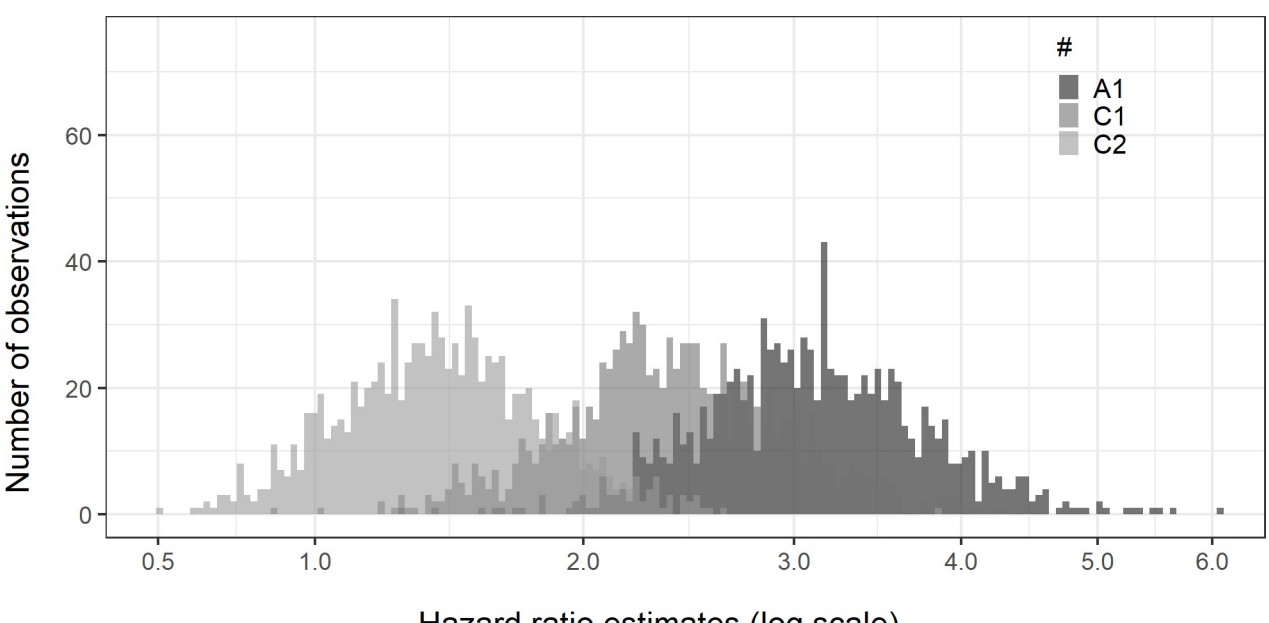

**Fig 4. Discovery scenario histogram.**

**Table 9. Discovery scenario legend.**

| # | A1 | C1 | C2 |
|---|---|---|---|
| $p_1$ | 0.25 | 0.25 | 0.25 |
| $p_2$ | 0.95 | 0.95 | 0.95 |
| $p_3$ | 0.05 | 0.05 | 0.05 |
| $p_4$ | 0.25 | 0.25 | 0.25 |
| $\lambda$ | 3.5 | 1.2 | 0.5 |

$p_1$ = proportion of correct CTCL diagnosis; $p_2$ = proportion of first-line use of TCS; $p_3$ = proportion of AD patients switching to tacrolimus; $p_4$ = proportion of CTCL patients switching to tacrolimus; $\lambda$ = scale parameter of the Weibull distribution used in the time to re-evaluation variable. Log-transformed x-axis of the hazard ratio estimates

1.49, 6.53), respectively [10] which is comparable to the ratio between the median simulated incidence rates of 0.145/0.047 = 3.09 for CTCL. In comparison, the JOELLE study found smaller effect sizes for the Danish, Swedish, and Dutch cohorts with adjusted incidence rate ratios of 1.28 (95% CI: 0.63–2.63), 1.75 (95% CI: 0.94, 3.25), and 1.33 (95% CI: 0.50–3.52), respectively. This may be due to greater use of tacrolimus as first- and second-line treatment as simulated in the *max use scenario* or a better ability to diagnose CTCL.

The strengths of the current simulation study include the multiple setting approach where key parameters in the multistate model used for simulation were varied to induce specific settings of interest. The model-based simulation approach made it possible to evaluate sources and magnitude of reverse causation bias in simulated cohort data closely aligned to that of the JOELLE [10].

The limitations of the current simulation study include the lack of confounders in the simulation model that may bias the results. However, the crude and adjusted estimates in the JOELLE study were comparable, indicating that the magnitude of confounding is small. Although our simulation results show that reverse causation may explain the observed associations between tacrolimus and CTCL in the JOELLE study, this does not exclude a potential causal association between tacrolimus and CTCL, however small.

We have used data from the JOELLE study to provide a framework to quantify the potential reverse causation bias when comparing first- and second-line treatments with an overlap in symptoms between the indication for the drug and the studied outcome. In the current simulation study, we found substantial reverse causation bias in the simulated CTCL risk estimates for patients treated with topical tacrolimus vs. TCS. Reverse causation bias may thus result in a false positive association between the second-line treatment and the studied outcome. The simulation-based framework can be adapted to other studies comparing first- and second-line treatments with an overlap in symptoms between the indication for the drug and the studied outcome. It allows an *a priori* exploration of the impact of various potential scenarios and can quantify the potential reverse causation bias.

## Supporting information

**S1 Fig. Example simulation of the *age at onset of AD* aka. *time from birth to AD* variable.** (PDF)

**S2 Fig. Example simulation of the *age at death* aka. *time from birth to* censoring variable.** (PDF)

**S3 Fig. Example simulation of the *time from initiation of TCS treatment to switching treatment* variable.**
(PDF)

**S4 Fig. Example simulation of the *time between AD diagnosis and CTCL diagnosis* variable.**
(PDF)

**S1 File. R code used for the simulations, output dataset of each simulation, and R code used in analysis.**
(ZIP)

## Acknowledgments

We thank Ross S. Baird for his assistance in preparing the manuscript; Alexandre Abramavicus for helping with dermatological insights and simulation assumptions; RTI Health Solutions for information and feedback about the JOELLE study and reverse causation; and Metadata for preparing the simulations.

## Author Contributions

**Conceptualization:** Christian Bjerregård Øland, Lise Skov Ranch, Thomas Delvin, Henny Bang Jakobsen, Christian Bressen Pipper.

**Data curation:** Christian Bjerregård Øland, Christian Bressen Pipper.

**Formal analysis:** Christian Bjerregård Øland, Christian Bressen Pipper.

**Methodology:** Christian Bjerregård Øland, Lise Skov Ranch, Tea Skaaby, Thomas Delvin, Henny Bang Jakobsen, Christian Bressen Pipper.

**Supervision:** Tea Skaaby, Christian Bressen Pipper.

**Validation:** Christian Bjerregård Øland, Christian Bressen Pipper.

**Visualization:** Christian Bjerregård Øland.

**Writing – original draft:** Christian Bjerregård Øland, Lise Skov Ranch, Tea Skaaby, Thomas Delvin, Henny Bang Jakobsen, Christian Bressen Pipper.

**Writing – review & editing:** Christian Bjerregård Øland, Lise Skov Ranch, Tea Skaaby, Thomas Delvin, Henny Bang Jakobsen, Christian Bressen Pipper.

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
