## [Decision Letter · Decision Letter 0]

2 Apr 2024

PONE-D-24-05762Reverse causation bias: a simulation study comparing first- and second-line treatments with an overlap of symptoms between treatment indication and studied outcomePLOS ONE

Dear Dr. Øland,

Thank you for submitting your manuscript to PLOS ONE. After careful consideration, we feel that it has merit but does not fully meet PLOS ONE’s publication criteria as it currently stands. Therefore, we invite you to submit a revised version of the manuscript that addresses the points raised during the review process.

We look forward to receiving your revised manuscript.

Kind regards,

Hideo Kato

Academic Editor

PLOS ONE

“All authors are or have been employees of LEO Pharma A/S and may be shareholders of LEO Pharma A/S. LEO Pharma A/S owns and sells topical tacrolimus/Protopic

®.”

5. We noted in your submission details that a portion of your manuscript may have been presented or published elsewhere. [Abstract has been presented at ICPE 2023.] Please clarify whether this publication was peer-reviewed and formally published. If this work was previously peer-reviewed and published, in the cover letter please provide the reason that this work does not constitute dual publication and should be included in the current manuscript.

6. For studies involving third-party data, we encourage authors to share any data specific to their analyses that they can legally distribute. PLOS recognizes, however, that authors may be using third-party data they do not have the rights to share. When third-party data cannot be publicly shared, authors must provide all information necessary for interested researchers to apply to gain access to the data. (https://journals.plos.org/plosone/s/data-availability#loc-acceptable-data-access-restrictions)

a) A description of the data set and the third-party source

b) If applicable, verification of permission to use the data set

c) Confirmation of whether the authors received any special privileges in accessing the data that other researchers would not have

d) All necessary contact information others would need to apply to gain access to the data

Reviewers' comments:

Reviewer's Responses to Questions

**Comments to the Author**

1. Is the manuscript technically sound, and do the data support the conclusions?

Reviewer #1: Yes

Reviewer #2: Yes

2. Has the statistical analysis been performed appropriately and rigorously? 

Reviewer #1: Yes

Reviewer #2: Yes

3. Have the authors made all data underlying the findings in their manuscript fully available?

Reviewer #1: Yes

Reviewer #2: Yes

4. Is the manuscript presented in an intelligible fashion and written in standard English?

Reviewer #1: Yes

Reviewer #2: Yes

5. Review Comments to the Author

Reviewer #1: This manuscript is very well written and describes a fascinating modeling exercise that addresses an important clinical issue. I appreciated the opportunity to read this excellent work. The findings of this manuscript are important for treating atopic dermatitis and are relevant much more generally.

Reviewer #2: This study investigated the impact of reverse causation on the association between the use of topical tacrolimus and CTCL in a multinational, population-based study using TCS as comparator.

Fig 1 is not correct and the authors should correct it so that it is clearly visible.

6. PLOS authors have the option to publish the peer review history of their article (what does this mean?). If published, this will include your full peer review and any attached files.

Reviewer #1: **Yes: **Steven Feldman

Reviewer #2: No

---

## [Author Response · Author response to Decision Letter 0]

26 Apr 2024

We thank the reviewers for their time in reviewing the paper. For specific responses to comments we refer to our uploaded "Response to Reviewers" document.

---

## [Decision Letter · Decision Letter 1]

8 May 2024

Reverse causation bias: a simulation study comparing first- and second-line treatments with an overlap of symptoms between treatment indication and studied outcome

PONE-D-24-05762R1

Dear Dr. Øland,

We’re pleased to inform you that your manuscript has been judged scientifically suitable for publication and will be formally accepted for publication once it meets all outstanding technical requirements.

Kind regards,

Hideo Kato

Academic Editor

PLOS ONE

Additional Editor Comments (optional):

Reviewers' comments:

Reviewer's Responses to Questions

**Comments to the Author**

1. If the authors have adequately addressed your comments raised in a previous round of review and you feel that this manuscript is now acceptable for publication, you may indicate that here to bypass the “Comments to the Author” section, enter your conflict of interest statement in the “Confidential to Editor” section, and submit your "Accept" recommendation.

Reviewer #2: All comments have been addressed

2. Is the manuscript technically sound, and do the data support the conclusions?

Reviewer #2: Yes

3. Has the statistical analysis been performed appropriately and rigorously? 

Reviewer #2: Yes

4. Have the authors made all data underlying the findings in their manuscript fully available?

Reviewer #2: Yes

5. Is the manuscript presented in an intelligible fashion and written in standard English?

Reviewer #2: Yes

6. Review Comments to the Author

Reviewer #2: *************************************************************************

The authors revise clearly.

7. PLOS authors have the option to publish the peer review history of their article (what does this mean?). If published, this will include your full peer review and any attached files.

Reviewer #2: No

---

## [Editor Report · Acceptance letter]

3 Jul 2024

PONE-D-24-05762R1 

PLOS ONE

Dear Dr. Øland, 

I'm pleased to inform you that your manuscript has been deemed suitable for publication in PLOS ONE. Congratulations! Your manuscript is now being handed over to our production team.

Kind regards, 

on behalf of

Dr. Hideo Kato 

Academic Editor

PLOS ONE